# Predictors for inpatient mortality during the first wave of the SARS-CoV-2 pandemic: A retrospective analysis

Daniel Sammartino[1]*, Farrukh Jafri[1], Brennan Cook[2], Lisa La[1], Hyemin Kim[1], John Cardasis[1], Joshua Raff[1]

1 White Plains Hospital, White Plains, New York, United States of America, 2 Rutgers Robert Wood Johnson School of Medicine, New Brunswick, New Jersey, United States of America

These authors contributed equally to this work.
* dsammartin@wphospital.org

**Data Availability Statement:** All relevant data are within the manuscript and its Supporting Information files.

## Abstract

### Background

The coronavirus disease 2019 (COVID-19) pandemic overwhelmed healthcare systems, highlighting the need to better understand predictors of mortality and the impact of medical interventions.

### Methods

This retrospective cohort study examined data from every patient who tested positive for COVID-19 and was admitted to White Plains Hospital between March 9, 2020, and June 3, 2020. We used binomial logistic regression to analyze data for all patients, and propensity score matching for those treated with hydroxychloroquine and convalescent plasma (CP). The primary outcome of interest was inpatient mortality.

### Results

1,108 admitted patients with COVID-19 were available for analysis, of which 124 (11.2%) were excluded due to incomplete data. Of the 984 patients included, 225 (22.9%) died. Risk for death decreased for each day later a patient was admitted [OR 0.970, CI 0.955 to 0.985; p < 0.001]. Elevated initial C-reactive protein (CRP) value was associated with a higher risk for death at 96 hours [OR 1.007, 1.002 to 1.012; p = 0.006]. Hydroxychloroquine and CP administration were each associated with increased mortality [OR 3.4, CI 1.614 to 7.396; p = 0.002, OR 2.8560, CI 1.361 to 6.160; p = 0.006 respectively].

### Conclusions

Elevated CRP carried significant odds of early death. Hydroxychloroquine and CP were each associated with higher risk for death, although CP was without titers and was adminis-tered at a median of five days from admission. Randomized or controlled studies will better describe the impact of CP. Mortality decreased as the pandemic progressed, suggesting

**Funding:** The authors received no specific funding for this work.

**Competing interests:** The authors have declared that no competing interests exist.

that institutional capacity for dynamic evaluation of process and outcome measures may benefit COVID-19 survival.

## Introduction

The coronavirus disease 2019 (COVID-19) was first diagnosed in the United States in January 2020 and has quickly become a public health emergency [1]. Early in the pandemic, the New York metropolitan area emerged as the epicenter of the global crisis from March through June 2020, accounting for 30% of all cases in the United States as of April 2020 [2, 3]. White Plains Hospital, which has the busiest Emergency Department in Westchester County increased overall capacity by 150% and critical care capacity by 500%. At the peak of the crisis, the hospital was at 79.6% of surge capacity.

During the first wave of the pandemic, with elevated inpatient mortality rates, a search for effective therapeutics was broadly launched. Hydroxychloroquine was initially utilized based upon reported clinical benefit either alone or in combination with azithromycin [4, 5]. Also emerging were reports of CP as a means of antibody transfer. This process had been used for the Spanish Influenza, H5N1 avian influenza, and H1N1 influenza [6, 7]. With this prior experience, the use of CP was initiated locally on April 9th, under an emergency Investigational New Drug (IND) application. Subsequently, plasma was administered under a national expanded access protocol (EAP) from April 11th through June 9th.

The objective of this report is to define associations between baseline health characteristics, severity of disease indices, as well as the impact of hydroxychloroquine and CP on inpatient mortality during the first wave of this pandemic amongst a specific cohort in Westchester County.

## Methods

### Study design/setting/participants

This retrospective, observational, cohort study included all patients over 18 years old with an initial positive COVID-19 polymerase chain reaction (PCR) test admitted to White Plains Hospital during its first defined wave from March 9, 2020, through June 3, 2020. White Plains Hospital is a 292-bed not-for-profit community hospital and member of the Montefiore Health System, located in the city of White Plains, NY. One-thousand one-hundred seventy-four patient records were accessed for those who were hospitalized with COVID-19. The White Plains Hospital Institutional Review Board (WPH IRB) approved this study and waived the requirement for informed consents.

### Data collection

Prior to a data collection plan, a directed acyclic graph (DAG) was developed to visualize and better understand potential confounders on admitted COVID-19 patients and risk of inpatient mortality (Fig 1). DAGs provide a simple way to graphically represent key concepts of relevance to researchers and help delineate potential confounders. Once a draft DAG was created, it was subsequently viewed by all members of the team until group consensus was obtained. The final version included categories of patient demographics, baseline health conditions, interventions offered during admission along with predictors of disease severity. Clinical data was then extracted and stored in REDCap based upon the findings of the DAG [8].

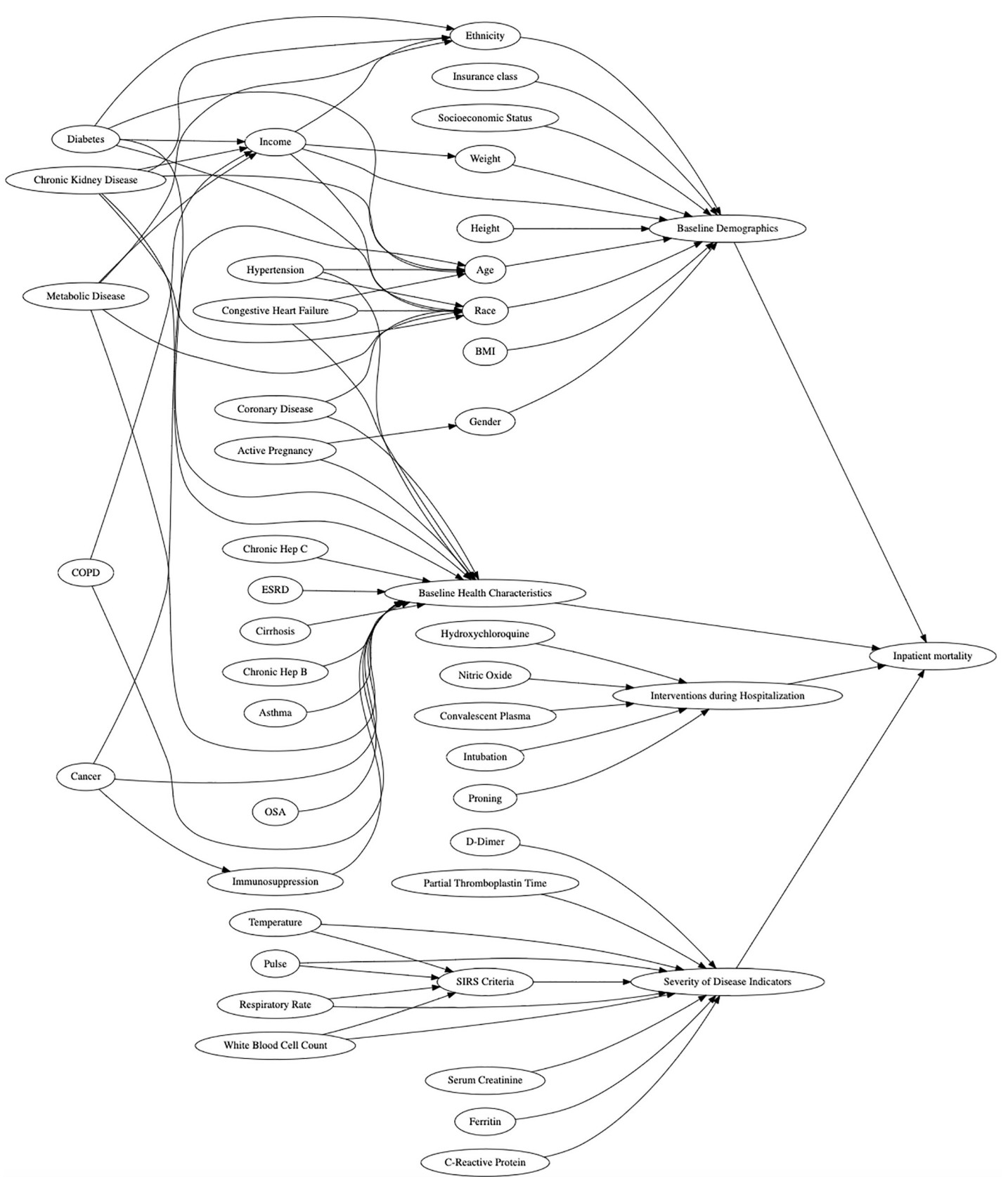

**Fig 1. Directed acrylic graph (DAG).** (A) Reviewing potential relationships associated with inpatient mortality. Key elements detected by the authors included baseline demographics, baseline health characteristics, interventions during hospitalization and severity of disease indicators.

Laboratory, baseline health, demographic and medication data were queried from Meditech using Microsoft SQL Server Data Tools (version 2017). COVID-19 patient encounters documented in REDCap for clinical research were utilized as the primary dataset for data extraction. Extraction focused on initial laboratory findings collected on day of admission, either in the ED or in the inpatient unit, if not collected in the ED. Medication orders placed towards those encounters were transferred to a spreadsheet. Sequential numbers were used to mask a patient's actual Meditech encounter information. Body mass index (BMI) was calculated using the weight in kilograms divided by the height squared in meters.

Patient zip code was used to estimate median income and per capita income using https://www.incomebyzipcode.com/. Insurance class was categorized as one of the following: Commercial, Medicaid, Medicare or Self-Pay.

## Collection and distribution of convalescent plasma

CP data was obtained through the New York Blood Center and the American Red Cross as these were the suppliers contracted with White Plains Hospital. Antibody titers were not performed prior to administration of CP as commercial assays were not readily available. Patients receiving CP met conditions for administration in the Mayo Clinic Expanded Access Program (EAP) protocol and those patients were all first enrolled into the EAP. Patients were selected for plasma administration at the discretion of the treating physician. Patients enrolled prior to May 1, 2020, were administered one unit of CP while patients enrolled beginning on May 1, 2020, were administered two units of CP. This change occurred based upon modifications and clarifications of study protocol from the sponsor.

## Statistical analysis

Of the 1,174 COVID-19 related hospitalizations, 1,108 were unique patients and 66 were readmissions. Readmitted patients were excluded from analysis, in order to prevent bias for patients that may have had previous antibodies to COVID-19.

We compared characteristics of COVID-19 positive patients according to baseline demographics, interventions offered, and traditional predictors of disease severity.

While controlling for selected covariates, logistic regression was used to model the parameter of interest, odds ratio (OR) of inpatient mortality. The covariates identified as confounders by the DAG were included in the logistic regression model. The covariate selection procedures investigated in this study utilized a total of four models to provide a robust understanding of the associations between predictors of disease severity and interventions offered with inpatient mortality.

All statistical analyses were performed in R version 4.0.2 (R Core Team). Mortality was assessed using binomial logistic regression, controlling for three subsets of demographic variables. Including all available variables resulted in overfitting due to collinearity as determined by variance inflation factors ("vif" function from the car package in R). Univariate logistic regression (Table 1) was then performed on each independent variable, and variables with a p value of less than 0.25 (Hosmer DW, Lemeshow S, Sturdivant) were selected for inclusion in a multinomial logistic regression termed Model 1a. Patients with missing data were omitted from the models. Multicollinearity was evaluated using the "alias" function from the stats

**Table 1. Univariate analysis.**

| | Odds Ratio | 2.5% CI | 97.5% CI | P Value |
|---|---|---|---|---|
| **Demographics** | | | | |
| Sex | 0.978 | 0.734 | 1.305 | 0.877 |
| Age | 1.043 | 1.033 | 1.053 | <0.001 |
| Estimated Per Capita Income | 1.000 | 1.000 | 1.000 | 0.030 |
| BMI | 0.992 | 0.971 | 1.011 | 0.432 |
| Admission Date | 0.984 | 0.975 | 0.994 | 0.001 |
| Total SIRS Score on Admission | 1.296 | 1.123 | 1.499 | <0.001 |
| **Insurance** | | | | |
| Medicaid | 1.095 | 0.664 | 1.790 | 0.719 |
| Medicare | 2.617 | 1.826 | 3.815 | <0.001 |
| Self Pay | 1.083 | 0.164 | 4.209 | 0.920 |
| **Race** | | | | |
| American Indian/Alaskan Native | 0.753 | 0.038 | 5.139 | 0.801 |
| Asian | 1.205 | 0.424 | 3.022 | 0.704 |
| Black | 0.705 | 0.473 | 1.032 | 0.078 |
| Other | 0.771 | 0.370 | 1.481 | 0.457 |
| Unknown | 0.789 | 0.510 | 1.194 | 0.274 |
| **Ethnicity** | | | | |
| Hispanic | 0.989 | 0.693 | 1.394 | 0.951 |
| Unknown | 1.644 | 0.897 | 2.913 | 0.096 |
| **Medical History** | | | | |
| Cancer | 0.994 | 0.324 | 2.542 | 0.990 |
| Hypertension | 1.903 | 1.420 | 2.566 | <0.001 |
| Coronary Artery Disease | 2.208 | 1.513 | 3.198 | <0.001 |
| Congestive Heart Failure | 2.690 | 1.572 | 4.551 | <0.001 |
| Asthma | 0.516 | 0.235 | 1.006 | 0.071 |
| Chronic Obstructive Pulmonary Disease | 1.969 | 1.189 | 3.200 | 0.007 |
| Chronic Renal Insufficiency | 2.372 | 1.450 | 3.831 | <0.001 |
| Hepatitis C | 2.554 | 0.500 | 11.660 | 0.222 |
| Cirrhosis | 3.429 | 0.947 | 12.424 | 0.053 |
| Diabetes | 1.236 | 0.912 | 1.668 | 0.168 |
| **Treatments** | | | | |
| Received Convalescent Plasma | 3.880 | 2.575 | 5.845 | <0.001 |
| Received Convalescent Plasma During Hemodialysis | 4.292 | 1.127 | 17.457 | 0.031 |
| Received Convalescent Plasma While Intubated | 7.443 | 3.830 | 15.171 | <0.001 |
| Placed in a Prone Position | 2.820 | 2.010 | 3.946 | <0.001 |
| Received Hydroxychloroquine | 2.644 | 1.974 | 3.546 | <0.001 |
| **Lab Values** | | | | |
| Ferritin | 1.001 | 1.000 | 1.001 | <0.001 |
| C Reactive Protein | 1.008 | 1.006 | 1.010 | <0.001 |
| D Dimer | 1.000 | 1.000 | 1.000 | <0.001 |

Odds ratios were determined through binomial regression and are presented with 95% CI (confidence intervals). Variables controlled for in the models are included in this table. For race, odds ratios were calculated against those self reporting as Caucasian, and for insurance, those with commercial insurance.

package in R. Given that many patients did not have labs drawn for CRP, Ferritin, or D-Dimer, Model 1b was a multinomial logistic regression which omitted these variables from the model to increase patient sample size.

Our team also wanted to evaluate mortality within 96 hours of admission. This stemmed from the belief that laboratory values and SIRS criteria in the ED may better control for severity of disease as it related to mortality earlier in the hospitalization. Model 2a was a multiple binomial logistic regression which evaluated mortality within 96 hours of admission. Variables were selected on a basis of univariate analysis again, and patients with missing observations were omitted from the final model. Model 2b was also a multiple binomial logistic regression which evaluated mortality within 96 hours, but omitted ferritin and d-dimer to increase sample size.

Nearest neighbor propensity score matching was applied on a 1:1 ratio to understand the treatment effect of Hydroxychloroquine and CP—the two potential therapeutics at our hospital at the time—with respect to inpatient mortality [9]. The demographic variables used in Models 1a and 1b to create a matched dataset including all patients who received hydroxychloroquine, and an equal number of patients who did not. Categorical variables included in the matched cohorts were compared using a chi square test and continuous variables were first evaluated with a Shapiro-Wilk's test and then a Wilcoxon test. A univariate logistic regression based on the matched cohorts was used to evaluate mortality and generated Models 3a and 3b. Within model 3a, the two groups differed in median ferritin and CRP levels as determined by a Wilcoxon test, but were more similar than the unmatched cohorts (ferritin, unmatched, $p = 1.16 \times 10^{-6}$ vs. matched $p = 0.007$, and CRP, unmatched $p = 6.45 \times 10^{-10}$, vs. matched $p = 0.037$). The same methodology was repeated for Models 4a and 4b with respect to the CP cohort.

Variance inflation factors were calculated for all multinomial logistic regression analyses and models were also evaluated using the Hosmer-Lemeshow goodness of fit.

## Results

### Characteristics of admitted patients

Amongst the 1,174 patients admitted to White Plains Hospital during the study time period, sixty-six patients were readmitted a second time, and three admitted for a third visit. Among this group of sixty-six individuals, thirteen died. All readmissions were excluded from our analysis.

During the study period, 1,108 unique patients were admitted for COVID-19. Data was excluded due to missing or incomplete data, and this varied based upon independent variables selected for each model. Characteristics for admitted patients across each model are available in Fig 2 for comparison.

### Characteristics and factors associated with hospital death

Model 1a and 1b both demonstrated an increased risk of mortality associated with increasing age [OR 1.049, CI 1.011 to 1.090 and OR 1.074, CI 1.053 to 1.095 respectively], earlier date of admission [OR 0.989, CI 0.957 to 1.021 and OR 0.972, CI 0.958 to 0.987], hydroxychloroquine administration [OR 3.142, CI 1.358 to 7.585 and OR 3.011, CI 1.962 to 4.67 respectively], CP administration [OR 4.216, CI 1.626 to 11.344 and OR 3.797, CI 2.019 to 7.231 respectively], and hemodialysis [OR 13.0, CI 2.932 to 71.726 and OR 7.029, CI 3.34 to 15.289 respectively] (Fig 3).

Model 1a included lab values for CRP and ferritin which showed increasing levels of both correlated with increased odds of inpatient mortality [OR 1.006, CI 1.002 to 1.010 and OR 1.001, CI 1.000 to 1.002, respectively]. Model 1a also had an increased odds of mortality for those utilizing Medicaid compared to Commercial insurance [OR 3.996, CI 1.315 to 12.877], however this trend did not persist with the greater sample size in Model 1b. Model 1b had a

| Demographics | Model 1a(N=328) | Model 1b (N=992) | Model 2a (N=490) | Model 2b (N=644) |
|---|---|---|---|---|
| Male sex - no. (%) | 182 (55.5) | 570 (57.5) | 304 (62) | 386 (59.9) |
| Median Age, Range | 66 (23-97) | 66.5 (17-104) | 65 (22-97) | 65 (19-98) |
| Median Body Mass Index (BMI), Range | 28.5 (14.16-69.6) | 23.2 (12.82-135.1) | 28.6 (14.16-71.63) | 28.8 (14.6 - 71.63) |
| Per Capita Income, By Zip Code (Median) | 47604 | 47604 | 47604 | 47604 |
| Median SIRS on Admission | 2 | 2 | 2 | 2 |
| **Race- no. (%)** | | | | |
| American Indian/Alaskan Native | 2(0.6) | 5(0.5) | 3(0.6) | 3 (0.5) |
| Asian | 5(1.5) | 21(2.1) | 11(2.2) | 17 (2.6) |
| Black | 63(19.2) | 196(19.8) | 85(17.3) | 122 (18.9) |
| Other | 19(5.8) | 48(4.8) | 28(5.7) | 35 (5.4) |
| Unknown | 59(17.9) | 149(15.0) | 88(17.9) | 104 (16.1) |
| White | 180(54.9) | 570(57.5) | 272(55.5) | 359 (55.7) |
| **Ethnicity- no. (%)** | | | | |
| Hispanic | 83(25.3) | 220(22.2) | 123(25.1) | 155 (24.07) |
| Not Hispanic | 228(69.5) | 719(72.5) | 337(68.8) | 455 (70.65) |
| Unknown | 17(5.2) | 53(5.4) | 30(6.1) | 34 (5.28) |
| **Insurance- no. (%)** | | | | |
| Commercial | 91 (27.7) | 279 (28.1) | 143 (29.2) | 194 (30.1) |
| Medicaid | 74 (22.6) | 192 (19.4) | 113 (23.1) | 138 (21.4) |
| Medicare | 159 (48.5) | 509 (51.3) | 228 (46.5) | 304 (47.2) |
| Self Pay / Other | 4(1.2) | 12(1.2) | 6(1.2) | 8 (1.2) |
| **Concomitant Treatments- no. (%)** | | | | |
| Convalescent Plasma | 61(18.8) | 105(10.6) | 97(19.8) | 104 (16.2) |
| Hydroxychloroquine | 137 (41.8) | 348 (35.1) | 227 (46.3) | 281 (43.6) |
| Placed in a Prone Position | 78(23.8) | 179(18.1) | 134(27.3) | 153 (23.8) |
| **Coexisting Medical Conditions - no.(%)** | | | | |
| Cancer | 49(14.9) | 152(15.4) | 70(14.3) | 88 (13.7) |
| Asthma | 25 (7.6) | 63 (6.4) | 30 (6.1) | 44 (6.8) |
| COPD | 18 (5.5) | 74 (7.5) | 25 (5.1) | 34 (5.3) |
| Hepatitis C | 2 (0.6) | 7 (0.7) | 3 (0.6) | 4 (0.6) |
| Cirrhosis | 2 (0.6) | 10 (1) | 2 (0.4) | 4 (0.6) |
| Chronic Renal Insufficiency | 19(5.85) | 70(7.1) | 34(6.9) | 45 (7.0) |
| Congestive Heart Failure | 8(2.46) | 58(5.8) | 19(3.9) | 27 (4.2) |
| Coronary Artery Disease | 37 (11.3) | 135 (13.6) | 64 (13.1) | 84 (13) |
| Diabetes | 112 (34.1) | 316 (31.9) | 156 (31.8) | 212 (32.9) |
| Hypertension | 182 (55.5) | 531 (53.5) | 269 (54.9) | 351 (54.5) |
| **Laboratories upon Admission (Median)** | | | | |
| CRP | 129.5 (0.3 - 450) | | 142 (0.3 - 495) | |
| D-Dimer | 1232 (229 - 123862) | | 1342.5 (229 - 123862) | |
| Ferritin | 573 (23 - 109686) | | 575 (23 - 109686) | |

**Fig 2. Demographics and clinical characteristics of the patients at baseline.** (A) Model 1a and 1b are binomial logistic regression models with respect to mortality. Lab values were included in 1a, but not 1b. Model 2a and 2b are binomial logistic regression models with respect to mortality within 96 hours of admission. Again, lab values were included in 2a, but not 2b. Demographic information presented represents individuals included in the specified model. Percentages are denoted within parenthesis. Per capita income was determined by patient's zip code using incomebyzipcode.com. The systemic inflammatory response syndrome score (SIRS) was calculated for hospital admission using the initial white blood cell count, temperature, pulse and respiratory rate present in the ED. Race and ethnicity were self-reported. Values differ due to individuals missing essential data being excluded from the models.

greater sample size than Model 1a and indicated an increased risk of mortality for patients with cirrhosis [OR: 6.542, CI 1.229 to 32.968] and congestive heart failure [OR 2.428, CI 1.173 to 5.027].

Similar trends were evident in Models 2a and 2b, both of which analyzed mortality within 96 hours of admission. Model 2b, which offered the largest sample size for those who had a complete CRP lab value, indicated increasing CRP increased the odds of mortality [OR 1.007, CI 1.002 to 1.012].

| Demographics | Model 1a(N=328) | Model 1b (N=992) | Model 2a (N=490) | Model 2b (N=644) |
|---|---|---|---|---|
| Deaths - no.(%) | 78(23.8) | 229(23.1) | 11(2.2) | 18(2.8) |
| Male sex | - | - | 0.738 CI (0.184-2.927) p = 0.66 | 0.601 CI (0.199-1.787) p = 0.36 |
| Age | 1.049 CI (1.011-1.09) p = 0.012 | 1.07 CI (1.05-1.091) p = <0.001 | 1.037 CI (0.967-1.115) p = 0.31 | 1.04 CI (0.986-1.1) p = 0.16 |
| Median Body Mass Index (BMI) | - | - | 0.967 CI (0.853-1.058) p = 0.53 | 0.982 CI (0.891-1.056) p = 0.66 |
| Per Capita Income | 1.000003 CI (0.99998 - 1.00002) p = 0.75 | 1.000003 CI (0.999995 - 1.00001) p = 0.483 | - | - |
| Date of Admission | 0.989 CI (0.957-1.021) p = 0.51 | 0.972 CI (0.958-0.987) p = < 0.001 | - | - |
| SIRS on Admission | 0.566 CI (0.293-1.07) p = 0.08 | 1.036 CI (0.759-1.415) p = 0.82 | 0.712 CI (0.299-1.603) p = 0.42 | 0.592 CI (0.296-1.133) p = 0.12 |
| **Race- no. (%)** | | | | |
| American Indian/Alaskan Native | - | 0.718 CI (0.023-8.337) p = 0.81 | - | - |
| Asian | 1.086 CI (0.063-26.147) p = 0.96 | 1.827 CI (0.547-5.683) p = 0.31 | - | - |
| Black | 0.826 CI (0.282-2.296) p = 0.72 | 0.933 CI (0.557-1.543) p = 0.79 | - | - |
| Other | 0.847 CI (0.173-3.583) p = 0.83 | 1.126 CI (0.41-2.88) p = 0.81 | - | - |
| Unknown | 1.975 CI (0.717-5.421) p = 0.19 | 1.239 CI (0.698-2.166) p = 0.46 | - | - |
| **Insurance- no. (%)** | | | | |
| Medicaid | 3.929 CI (1.3-12.571) p = 0.017 | 1.715 CI (0.914-3.215) p = 0.09 | 1.339 CI (0.04-40.003) p = 0.85 | 1.847 CI (0.202-16.908) p = 0.56 |
| Medicare | 1.792 CI (0.626-5.477) p = 0.29 | 0.985 CI (0.561-1.737) p = 0.96 | 2.009 CI (0.199-49.849) p = 0.59 | 1.191 CI (0.207-9.834) p = 0.85 |
| Self Pay / Other | - | 1.159 CI (0.151-6.064) p = 0.87 | - | - |
| **Concomitant Treatments- no. (%)** | | | | |
| Convalescent Plasma | 3.233 CI (1.387-7.603) p = 0.007 | 2.935 CI (1.682-5.13) p= < 0.001 | 0.314 CI (0.016-1.96) p = 0.29 | 0.304 CI (0.04-1.368) p = 0.17 |
| Hydroxychloroquine | 3.131 CI (1.383-7.411) p = 0.007 | 2.737 CI (1.812-4.169) p= < 0.001 | 0.9 CI (0.204-3.539) p = 0.88 | 0.592 CI (0.172-1.798) p = 0.37 |
| Placed in a Prone Position | 2.764 CI (1.245-6.186) p = 0.012 | 3.628 CI (2.283-5.802) p= < 0.001 | - | - |
| **Coexisting Medical Conditions - no.(%)** | | | | |
| Cancer | 1.808 CI (0.73-4.39) p = 0.19 | 1.369 CI (0.851-2.181) p = 0.19 | - | - |
| Asthma | 1.213 CI (0.148-6.375) p = 0.84 | 0.786 CI (0.323-1.753) p = 0.57 | - | - |
| COPD | 0.716 CI (0.146-2.822) p = 0.65 | 1.294 CI (0.688-2.389) p = 0.42 | 1.16 CI (0.051-8.892) p = 0.90 | 2.395 CI (0.457-9.631) p = 0.25 |
| Hepatitis C | - | 2.795 CI (0.283-22.927) p = 0.35 | - | - |
| Cirrhosis | - | 6.157 CI (1.164-31.857) p = 0.029 | - | - |
| Chronic Renal Insufficiency | 2.801 CI (0.748-10.336) p = 0.12 | 1.515 CI (0.807-2.809) p = 0.19 | 2.112 CI (0.247-12.905) p = 0.44 | 2.352 CI (0.521-9.142) p = 0.23 |
| Congestive Heart Failure | 5.065 CI (0.485-44.025) p = 0.15 | 2.499 CI (1.228-5.093) p = 0.01 | 1.523 CI (0.151-10.82) p = 0.69 | 0.9 CI (0.112-4.615) p = 0.91 |
| Coronary Artery Disease | 0.94 CI (0.309-2.705) p = 0.91 | 1.37 CI (0.824-2.264) p = 0.22 | 2.18 CI (0.476-9.671) p = 0.30 | 2.143 CI (0.637-6.939) p = 0.21 |
| Diabetes | 0.994 CI (0.466-2.089) p = 0.99 | 0.826 CI (0.545-1.243) p = 0.36 | 1.692 CI (0.214-35.947) p = 0.66 | 1.895 CI (0.404-13.926) p = 0.46 |
| Hypertension | 1.886 CI (0.841-4.378) p = 0.13 | 1.229 CI (0.823-1.841) p = 0.31 | 0.944 CI (0.209-5.165) p = 0.94 | 0.818 CI (0.252-2.946) p = 0.74 |
| **Laboratories upon Admission (Median)** | | | | |
| CRP | 1.005 CI (1.001-1.01) p = 0.01 | - | 1.004 CI (0.997-1.01) p = 0.26 | 1.006 CI (1.001-1.01) p = 0.015 |
| D-Dimer | 1.00001 CI (0.99999 - 1.00004) p=0.39 | - | 1.000005 CI (0.99995 - 1.00004) p =0.79 | - |
| Ferritin | 1.001 CI (1-1.002) p = 0.055 | - | - | - |

**Fig 3. Odds ratios for inpatient mortality.** (A) Odds ratios for mortality were determined through binomial logistic regression and are presented with 95 percent confidence intervals. Variables controlled for in the models are included in this table. For race, odds ratios were calculated against those self-reporting as Caucasian, and for insurance, those with commercial insurance.

## Hydroxychloroquine

Both multiple binomial logistic regression analyses saw an increased odds of mortality for those who received hydroxychloroquine. When propensity score matching was used to create two sub cohorts, one matching demographic variables in model 1a and the other matching variables included in model 1b, both cohorts saw an increased odds of death when analyzed with logistic regression [OR 3.40, CI 1.61 to 7.40 and OR 1.63, CI 1.08 to 2.45] (Fig 4). When propensity score matching using the demographic variables from Model 1a and subsequent binomial logistic regression was repeated for all patients who presented with a low SIRS score (0–1), those treated with hydroxychloroquine had an increased risk of mortality compared to those who were not given hydroxychloroquine (OR 4.23, CI 1.633 to 12.07, n = 94). When this method was repeated for those who had a high SIRS score (> = 2), a non-statistical trend towards increased mortality (OR 1.950, CI 0.982 to 3.940, n = 176) emerged.

## Convalescent plasma

Between April 9 and June 9, 2020, 117 patients received CP therapy at a median five days from admission. Antibody titers for plasma donors were not performed due limitations in reliable testing. Eighty-seven patients (77%) received one unit of CP. Beginning on May 1st, 26 patients (23%) received two units of CP. Binomial logistic regression determined that CP administration was associated with increased mortality in a set of 103 plasma recipients and in a

| Demographics | Model 3a - HCQ (N=137) | Model 3a - No HCQ (N=137) | Model 3b - HCQ (N=348) | Model 3b - No HCQ (N=348) | Model 4a - CP (N=61) | Model 4a - No CP (N=61) | Model 4b - CP (N=105) | Model 4b - No CP(N=105) |
|---|---|---|---|---|---|---|---|---|
| Deaths - no.(%) | 49 (35.8) | 23 (16.8) | 121 (34.8) | 66 (19) | 32 (52.5) | 16 (26.2) | 32 (30.5) | 38 (36.2) |
| Male sex | 83 (60.6) | 71 (51.8) | 222 (63.8) | 196 (56.3) | 38 (62.3) | 32 (52.5) | 38 (36.2) | 64 (61) |
| Median Age | 63 (23 - 96) | 66 (31 - 96) | 63 (19 - 97) | 65 (18 - 98) | 67 (41 - 96) | 70 (32 - 97) | 67 (41 - 96) | 70 (32 - 97) |
| Median Body Mass Index (BMI) | 30.4 (19.9 - 53.3) | 28.1 (18.0 - 69.6) | 29.4 (17.0 - 71.6) | 28.1 (13.9 - 69.6) | 29.6 (19.9 - 52.8) | 28.2 (14.2 - 69.6) | 29.7 (19.9 - 52.8) | 28.2 (14.2 - 69.6) |
| Per Capita Income (Median) | 47604 | 45471 | 47604 | 47604 | 44761 | 45471 | 44761 | 45471 |
| Median Length of Hospitalization (Days) | 10 (1 - 65) | 7 (1 - 44) | 9 (1 - 65) | 5 (1 - 44) | 18 (4 - 65) | 9 (1 - 38) | 18 (4 - 65) | 9 (1 - 38) |
| Date of Admission (Median) | 4/6/2020 | 4/12/2020 | 4/4/2020 | 4/5/2020 | 4/16/2020 | 4/21/2020 | 4/16/2020 | 4/13/2020 |
| Median SIRS on Admission | 2 | 2 | 2 | 2 | 2 | 2 | 2 | 2 |
| **Race- no. (%)** | | | | | | | | |
| American Indian/Alaskan Native | 0 | 0 | 2 (0.6) | 1 (0.3) | 1 (1.6) | 1 (1.6) | 1 (1) | 2 (1.9) |
| Asian | 3 (2.2) | 2 (1.5) | 7 (2) | 9 (2.6) | 2 (3.3) | 1 (1.6) | 2 (1.9) | 3 (2.9) |
| Black | 28 (20.4) | 25 (18.2) | 71 (20.4) | 70 (20.1) | 8 (13.1) | 12 (19.7) | 8 (7.6) | 12 (11.4) |
| Other | 10 (7.3) | 8 (5.8) | 21 (6) | 16 (4.6) | 4 (6.6) | 3 (4.9) | 4 (3.8) | 4 (3.8) |
| Unknown | 20 (14.6) | 25 (18.2) | 54 (15.5) | 50 (14.4) | 15 (24.6) | 12 (19.7) | 15 (14.3) | 24 (22.9) |
| White | 76 (55.5) | 77 (56.2) | 193 (55.5) | 202 (58) | 31 (50.8) | 32 (52.5) | 31 (29.5) | 60 (57.1) |
| **Ethnicity- no. (%)** | | | | | | | | |
| Hispanic | 30 (21.9) | 41 (29.9) | 74 (21.3) | 84 (24.1) | 20 (32.8) | 12 (19.7) | 20 (19) | 26 (24.8) |
| Not Hispanic | 100 (73) | 90 (65.7) | 254 (73) | 245 (70.4) | 37 (60.7) | 44 (72.1) | 37 (35.2) | 71 (67.6) |
| Unknown | 7 (5.1) | 6 (4.4) | 20 (5.7) | 19 (5.5) | 4 (6.6) | 5 (8.2) | 4 (3.8) | 8 (7.6) |
| **Insurance- no. (%)** | | | | | | | | |
| Commercial | 44 (32.1) | 35 (25.5) | 118 (33.9) | 111 (31.9) | 17 (27.9) | 18 (29.5) | 17 (16.2) | 30 (28.6) |
| Medicaid | 34 (24.8) | 34 (24.8) | 79 (22.7) | 76 (21.8) | 13 (21.3) | 10 (16.4) | 13 (12.4) | 20 (19) |
| Medicare | 57 (41.6) | 66 (48.2) | 145 (41.7) | 158 (45.4) | 31 (50.8) | 33 (54.1) | 31 (29.5) | 55 (52.4) |
| Self Pay / Other | 2 (1.5) | 2 (1.5) | 6 (1.7) | 3 (0.9) | 0 | 0 | 0 | 0 |
| **Concomitant Treatments- no. (%)** | | | | | | | | |
| Convalescent Plasma | 28 (20.4) | 26 (19) | 52 (14.9) | 36 (10.3) | 61 (100) | 0 | 61 (58.1) | 0 |
| Hydroxychloroquine | 137 (100) | 0 | 348 (100) | 0 | 28 (45.9) | 23 (37.7) | 28 (26.7) | 49 (46.7) |
| Placed in a Prone Position | 48 (35) | 26 (19) | 106 (30.5) | 61 (17.5) | 23 (37.7) | 20 (32.8) | 23 (21.9) | 50 (47.6) |
| **Coexisting Medical Conditions - no.(%)** | | | | | | | | |
| Cancer | 17 (12.4) | 19 (13.9) | 48 (13.8) | 50 (14.4) | 8 (13.1) | 9 (14.8) | 8 (7.6) | 15 (14.3) |
| Asthma | 8 (5.8) | 13 (9.5) | 21 (6) | 23 (6.6) | 2 (3.3) | 1 (1.6) | 2 (1.9) | 2 (1.9) |
| COPD | 8 (5.8) | 9 (6.6) | 21 (6) | 22 (6.3) | 3 (4.9) | 3 (4.9) | 3 (2.9) | 6 (5.7) |
| Hepatitis C | 1 (0.7) | 0 | 3 (0.9) | 1 (0.3) | 1 (1.6) | 1 (1.6) | 1 (1) | 2 (1.9) |
| Cirrhosis | 0 | 0 | 0 | 0 | 1 (1.6) | 1 (1.6) | 1 (1) | 2 (1.9) |
| Chronic Renal Insufficiency | 9 (6.6) | 7 (5.1) | 27 (7.8) | 27 (7.8) | 3 (4.9) | 3 (4.9) | 3 (2.9) | 10 (9.5) |
| Congestive Heart Failure | 3 (2.2) | 4 (2.9) | 11 (3.2) | 11 (3.2) | 1 (1.6) | 1 (1.6) | 1 (1) | 6 (5.7) |
| Coronary Artery Disease | 17 (12.4) | 16 (11.7) | 42 (12.1) | 40 (11.5) | 8 (13.1) | 5 (8.2) | 8 (7.6) | 19 (18.1) |
| Diabetes | 53 (38.7) | 42 (30.7) | 118 (33.9) | 110 (31.6) | 23 (37.7) | 23 (37.7) | 23 (21.9) | 37 (35.2) |
| Hypertension | 82 (59.9) | 71 (51.8) | 180 (51.7) | 181 (52) | 35 (57.4) | 39 (63.9) | 35 (33.3) | 57 (54.3) |
| **Laboratories upon Admission (Median)** | | | | | | | | |
| CRP | 144.7 (0.3 - 434.6) | 122.3 (1.3 - 449.8) | | | 155.8 (7.6 - 434.6) | 151.6 (8.6 - 444.4) | | |
| D-Dimer | 1116 (236 - 123862) | 1234 (229 - 84843) | | | 1421 (273 - 106000) | 1881 (241 - 123862) | | |
| Ferritin | 704 (50 - 1866) | 541 (23 - 3004) | | | 631 (72 - 1866) | 671 (55 - 3004) | | |

**Fig 4. Demographic data for propensitiy score matching of hydroxychloroquine (HCQ) and convalescent plasma (CP) groups.** (A) Propensity score matching was used to create a number of cohorts, and demographics for each cohort are shown in the table above. Model 3a and 3b separate groups with respect to hydroxychloroquine, and model 4a and 4b with respect to convalescent plasma. Percentages of each group are expressed in parenthesis for categorical variables and ranges for continuous. Race and ethnicity are self-reported. Models 3a and 4a included lab values for matching.

propensity score matching cohort in which 61 of these plasma recipients were matched with 61 patients who did not receive plasma [OR 2.86, CI 1.36 to 6.16; p = 0.005, N = 122].

## Discussion

In this retrospective analysis of 1,108 hospitalized COVID-19 patients at an epicenter during the first wave of the pandemic, we were able to identify risk factors and associations with mortality. Consistent with previous reports, we observed that increasing age and certain pre-existing medical conditions relating to major organ systems were associated with increased chance for death [10, 11].

Our data also reports a mortality odds ratio of 0.972 for each day later a person was admitted, suggesting that for every week or month later that a person was admitted, their risk of death dropped by 16% and 49%, respectively. We found this information of interest given the lack of efficacy, and even an association of harm, amongst our main medical therapeutics. To account for this finding, we surmise that the improved survival may be explained by multiple contributing factors related to institutional and supportive measures.

Structure and process measures in our COVID-19 response were dynamically evaluated with changes implemented during the first wave as organizational leadership actively applied the Donabedian model towards our response [12]. Initially during the crisis, intubation was used readily to manage severely hypoxic patients along with those in rapid respiratory decline.

In April, the hospital developed and trained through medical simulation a highly skilled Critical Airway Team consisting of Anesthesiologists, Emergency Physicians, ICU Physicians and Respiratory Therapists [13]. This team managed over 80% of the intubations in the hospital and developed a higher threshold for intubation across the institution with the goal of increased utilization of other modalities for hypoxic patients including high flow nasal cannula and BiPAP. Over time, we were able to introduce less invasive methods for respiration [14–16], leading to fewer intubations which may have led to an improved chance for survival. For patients requiring prolonged mechanical ventilation, we observed that converting to tracheostomy when feasible, was associated with improved clinical outcomes, such as decreased sedative use, earlier participation in physical therapy, improved odds of ventilator liberation, and better allocation of resources [17, 18]. The formation of specialized proning teams and tracheostomy teams was developed over the course of the surge, leading to more efficient procedural care.

Elevated presentation values for CRP (run on Siemens Advia XPT; normal range 0.0–9.9 mg/L) demonstrated a strong correlation with death within 96 hours of hospital admission. As an acute inflammatory protein produced by the liver, elevated CRP is felt to be an indicator of cytokine storm [19, 20], and this data further suggests that an elevated value upon presentation could portend impending organ system collapse [21]. Prospective studies can validate these findings as well as the potential benefit of interventions tailored to the clinical urgency of patients admitted with elevated CRP.

At the time of this initial surge, there was no hospital-wide policy for the use of hydroxychloroquine, and its prescription was left to the discretion of the treating medical team. In several of our models, the results suggest a trend towards increase in overall mortality in those patients treated with hydroxychloroquine. Although retrospective in nature and subject to the potential selection bias of treating sicker patients, our findings are consistent with randomized trials [22, 23].

The associated benefit of CP in COVID-19 has been difficult to determine, although reports of high-titer plasma therapy administered within 72 hours have demonstrated reduced mortality [24, 25]. And while neither a matched cohort study of 64 patients [26] nor a randomized study of 333 patients was able to demonstrate benefit, in both those studies the median time to receive convalescent plasma was seven and eight days, respectively [27].

Although many of our results regarding CP showed an increased association with mortality, this was not a randomized trial. Eligibility for the EAP was broad, and it was offered at the discretion of the attending physicians. Matching sets were constructed based upon baseline and admission data. Since many of our initial patients who received CP were critically ill and likely hospitalized for days or even weeks, the complexity of their illness was not likely to be fully captured by our matching, and any potential benefit of convalescent plasma could have been missed. The median time from diagnosis to receipt of CP was five days, likely too far out for a potential beneficial effect. Finally, commercially available testing for coronavirus antibody titers was not available at the time of these treatments.

We encountered several limitations in this study. The retrospective design prevented us from capturing all relevant data for logistic regression, which led to incomplete data sets for matching analyses. We did not have access to detailed nurse to patient ratio data which we believe can be a confounder as an intervention. We attempted to control for the severity of patients' illness by utilizing baseline SIRS score in the ED, although this did not reflect the extent of the evolving hospital course. Corticosteroids and anticoagulation were used in varied formulations and dosing patterns which prevented a meaningful retrospective analysis.

Notwithstanding these limitations, this study provides continued data on the use of CRP as a marker for rapid decline and death for COVID-19. Ideally, more randomized or prospective

studies with convalescent plasma for COVID-19 will be able to address the potential benefit while controlling for antibody titers, time to administration, and relevant predictors of disease severity. In the absence of significant therapeutics for COVID-19, the continued advancements in supplemental oxygen delivery, tracheostomy use, and strategic staffing may play important roles in the improvement of hospital survival rates. Institutions capable of dynamic assessment and response to the changing treatment standards of COVID-19 may fare better.

## Supporting information

**S1 Appendix.**
(DOCX)

**S1 File.**
(PDF)

**S2 File.**
(PDF)

**S3 File.**
(PDF)

## Acknowledgments

We thank James Peacock MD, Cheryl Frydman MD, Beth Bruderline, Andrea Weaver, Christine Kopec, and Kavitha Vel for their assistance in data collection; Alan Multz MD, Michael Palumbo MD, Linda Vandervoort and Karen Banoff DNP for administrative support; Anshul Kumar PhD and Matt Ploenzke PhD for statistical assistance and oversight.

## Author Contributions

**Conceptualization:** Daniel Sammartino, Farrukh Jafri, Lisa La, Hyemin Kim, John Cardasis.

**Data curation:** Daniel Sammartino, Brennan Cook, Lisa La, Hyemin Kim, Joshua Raff.

**Formal analysis:** Daniel Sammartino, Farrukh Jafri, Brennan Cook, Lisa La, Hyemin Kim, John Cardasis, Joshua Raff.

**Investigation:** Daniel Sammartino.

**Methodology:** Daniel Sammartino, Lisa La, Joshua Raff.

**Project administration:** Daniel Sammartino, Farrukh Jafri, Lisa La, John Cardasis, Joshua Raff.

**Supervision:** Farrukh Jafri, John Cardasis, Joshua Raff.

**Validation:** Brennan Cook, Joshua Raff.

**Visualization:** Joshua Raff.

**Writing – original draft:** Daniel Sammartino, Farrukh Jafri, Brennan Cook, Hyemin Kim, Joshua Raff.

**Writing – review & editing:** Daniel Sammartino, Farrukh Jafri, Brennan Cook, Hyemin Kim, Joshua Raff.

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
