## [Decision Letter · Decision Letter 0]

16 Mar 2021

PONE-D-21-01444

Predictors for Inpatient Mortality during the first wave of the SARS-CoV-2 pandemic: A Retrospective Analysis

PLOS ONE

Dear Dr. Sammartino,

Thank you for submitting your manuscript to PLOS ONE. After careful consideration, we feel that it has merit but does not fully meet PLOS ONE’s publication criteria as it currently stands. Therefore, we invite you to submit a revised version of the manuscript that addresses the points raised during the review process.

We look forward to receiving your revised manuscript.

Kind regards,

Aleksandar R. Zivkovic

Academic Editor

PLOS ONE

2. In your ethics statement in the manuscript and in the online submission form, please ensure that you have discussed whether all data/samples were fully anonymized before you accessed them and/or whether the IRB or ethics committee waived the requirement for informed consent.

If patients provided informed written consent to have data/samples from their medical records used in research, please include this information.

3. In the ethics statement in the manuscript and in the online submission form, please provide additional information about the patient records/samples used in your retrospective study, including the date range (month and year) during which patients' medical records/samples were accessed.

5. Please include captions for your Supporting Information files at the end of your manuscript, and update any in-text citations to match accordingly. Please see our Supporting Information guidelines for more information: http://journals.plos.org/plosone/s/supporting-information

Note: HTML markup is below. Please do not edit.]

Reviewers' comments:

Reviewer #1: The authors have conducted a single center retrospective analysis of 1108 unique patients with the aim of predicting mortality of inpatients, and evaluating various treatment methods from the "retrospectoscope."

They took an interesting approach of creating developing a directed acyclic graph to look for potential confounders. I believe this process needs to better defined and referenced. While the CODE data is helpful to some, it is not helpful for those who are not coders. Possibly a pictogram to show the DAG would be beneficial.

Although it is inferred that patients with missing data were removed from specific models, this should be explicitly stated.

If patients with missing data were included this needs to be described in detail. Furthermore in the appendix it is unclear whether biomarker tests were exclusively limited to values obtained in the ED or at anytime during their admission. If a lab value was obtained later in hospitalization and not in the ED was that value still considered?

A table showing the univariate analysis between discharged and died patients should be included. Here important considerations such as why continuous variables were treated with median and not mean but in the propensity data you used only t-test? meaning that you should state clearly if you tested continuous variables for normal distribution in order to decide to represent variables with mean (SD) or median (IQR) and then the appropriate statistical test. Also, lab values could have been categorized as in many other papers that have assessed predictors for in-hospital mortality ("A Novel Severity Score to Predict Inpatient Mortality in COVID-19 patients," Altschul et al. Sci Rep, 2020)

Your table 1 is confusing in the sense that in your methods for the "b" models you omit values that were not routinely tested to increase sample size yet these models still include ferritin, d-dimer and CRP levels. Can you please clarify this?

The CRP OR is 1.006-1.007, although is technically >1 I'd argue its prediction value with a very weak value compared to some of the other results.

Also you have decided to exclude corticosteroids and anti-coagulation treatments due to a variance in treatment regimens offered, however I would argue that including this data would further confirm as what i see is the major limitation of this study in that the treatments that were given were given to patients presenting with more severe COVID-19 illness creating the selection bias we are seeing with hydroxychloroquine and CP treatments. Therefore since severity of illness, since it was not controlled for is the defining characteristic that is predicting mortality in covid-19 patients.

The singular interesting finding in this study in which i agree with the authors conclusions is that treatments improved over time and understanding. And that survival improved over those months during the major surge in New York as we intubated less patients, had better treatments and improved our critical care paradigms.

Reviewer #3: The authors present the results of an analysis of 1108 patients admitted for COVID-19, perform a binomial logistic regression analysis and propensity score matching for those treated with hydroxychloroquine and convalescent plasma (CP) and the main conclusions of the article show that elevated CRP carried significant odds of early death. Hydroxychloroquine and PC were associated with an increased risk of death, even though CP had no titers and was administered within a median of five days of admission.

Recommendations

1) I suggest removal from the conclusions Randomized or controlled studies will better describe the impact of CP. Mortality decreased as the pandemic progressed, suggesting that the institutional capacity for dynamic dynamic assessment of process and outcome measures may benefit the survival of COVID-19. Since these conclusions do not follow from the results reported in the abstract.

2) As what is relevant in the results is Elevated CRP carried significant odds of early death. And hydroxychloroquine and PC were associated with an increased risk of death. I don't know what it looks like for the authors to change the title of the article.

3) The results are presented synthetically, but the two tables are very dense with a lot of information, I don't know how the authors feel about simplifying or splitting the tables into two. Or transfer the information to an appendix.

---

## [Author Response · Author response to Decision Letter 0]

21 Apr 2021

Please see the attached file "Response to Reviewers" for a detailed response.

---

## [Editor Report · Decision Letter 1]

23 Apr 2021

Predictors for inpatient mortality during the first wave of the SARS-CoV-2 pandemic: A Retrospective analysis

PONE-D-21-01444R1

Dear Dr. Sammartino,

We’re pleased to inform you that your manuscript has been judged scientifically suitable for publication and will be formally accepted for publication once it meets all outstanding technical requirements.

Kind regards,

Aleksandar R. Zivkovic

Academic Editor

PLOS ONE

---

## [Editor Report · Acceptance letter]

30 Apr 2021

PONE-D-21-01444R1 

Predictors for inpatient mortality during the first wave of the SARS-CoV-2 pandemic: A retrospective analysis 

Dear Dr. Sammartino:

I'm pleased to inform you that your manuscript has been deemed suitable for publication in PLOS ONE. Congratulations! Your manuscript is now with our production department. 

Kind regards, 

on behalf of

Dr. Aleksandar R. Zivkovic 

Academic Editor

PLOS ONE